# Tight Junctions of the Outer Blood Retina Barrier

**DOI:** 10.3390/ijms21010211

**Published:** 2019-12-27

**Authors:** Aisling Naylor, Alan Hopkins, Natalie Hudson, Matthew Campbell

**Affiliations:** Smurfit Institute of Genetics, Trinity College Dublin, Dublin 2, Ireland; aisling.naylor@ucdconnect.ie (A.N.); ahopkin@tcd.ie (A.H.); natalie.hudson@tcd.ie (N.H.)

**Keywords:** retinal pigment epithelium, retinopathy, tight junction

## Abstract

The outer blood retina barrier (oBRB) formed by the retinal pigment epithelium (RPE) is critical for maintaining retinal homeostasis. Critical to this modified neuro-epithelial barrier is the presence of the tight junction structure that is formed at the apical periphery of contacting cells. This tight junction complex mediates size-selective passive diffusion of solutes to and from the outer segments of the retina. Unlike other epithelial cells, the apical surface of the RPE is in direct contact with neural tissue and it is centrally involved in the daily phagocytosis of the effete tips of photoreceptor cells. While much is known about the intracellular trafficking of material within the RPE, less is known about the role of the tight junction complexes in health and diseased states. Here, we provide a succinct overview of the molecular composition of the RPE tight junction complex in addition to highlighting some of the most common retinopathies that involve a dysregulation of RPE integrity

## 1. Introduction

The blood retina barrier (BRB) is fundamental in establishing and maintaining a suitable environment for optimum retinal function [1]. While the inner blood retina barrier (iBRB) is composed of tight junctions (TJs) between retinal capillary endothelial cells, the so-called outer blood retina barrier (oBRB) is formed by the TJs between retinal pigment epithelial (RPE) cells. This oBRB acts to regulate and filter molecular movement of solutes and nutrients from the choroid to the sub-retinal space. Impairment of either of these barriers can lead to the accumulation of blood-borne proteins and other potentially toxic solutes within the retina [1] (differences between the oBRB and iBRB are summarised further in Table 1).

The RPE is composed of a single layer of epithelial cells separating the neural retinal from the underlying Bruch’s membrane and fenestrated choriocapillaris. TJs connecting neighbouring RPE cells block the movement of plasma components and toxic molecules into the retina as well as allowing for a controlled flow of fluid and solutes across an osmotic gradient from the choroidal vasculature to the outer retina [2]. The retina has the highest oxygen consumption per weight of any tissue in the body and the BRB (both outer and inner) is essential in providing a distinct and regulated source of nutrients to facilitate for this high metabolic rate [3]. TJs at both the iBRB and oBRB are complex, dynamic structures [4] and in the context of these barriers, the integrity of these TJs is crucial to sight.

Specifically, the oBRB allows for the survival of the photoreceptors (PRs) by supporting essential functions including filtering and transport of nutrients and photoreceptor outer segment (POS) phagocytosis. Other important functions of the RPE include absorption of out of focus and scattered light, retinal adhesion, and vitamin A transport and processing, and re-isomerisation of all-trans-retinal to 11-cis retinal, which is crucial for the visual cycle [2,5]. The RPE is essential for visual function, a failure of any one of these functions can lead to degeneration of the retina, loss of visual function, and blindness [2]. The RPE is highly polarized and can be divided into apical and basolateral sides. The apical surface is in direct contact with the POS and the basolateral side acts as a barrier in its interaction with the highly permeable and highly perfused choriocapillaris of the choroid [6].

TJs allow for high degrees of selectivity in paracellular barrier function in both the iBRB and the oBRB. TJs are unique assemblies of transmembrane proteins and peripheral cytoplasmic proteins. Transmembrane proteins include the claudins, the MARVEL (Mal and related proteins for vesicle trafficking and membrane link) family and junctional adhesion molecules (JAMs), which span the plasma membrane. Peripheral cytoplasmic proteins such as zonula occludens-1, (ZO-1), -2 (ZO-2) and -3 (ZO-3) anchor these transmembrane proteins to the cytoskeleton and are vital in the initial formation and distinct organization of TJs [7,8].

In principle, the inner and outer BRBs are performing inherently similar roles of paracellular diffusion regulation; however, the molecular composition varies considerably in these junctions. In this review, we will discuss the structure and functions of the oBRB, and how its disruption contributes to the pathogenesis of a variety of ocular conditions including diabetic retinopathy (DR), age related macular degeneration (AMD), central serous chorioretinopathy (CSCR), Sorsby’s fundus dystrophy, Retinitis Pigmentosa (RP), and conditions associated with mutations in CLDN-19, the gene encoding for the TJ protein claudin-19.

## 2. The Retinal Pigment Epithelium (RPE)

The RPE is composed of a single layer of cells joined laterally towards their apices by TJs between adjacent plasma membranes (Figure 1a demonstrates the topographic relationship of the RPE within the retina and Figure 1b demonstrates the structure and function of the RPE as described in this section). The apical membrane faces the photoreceptor outer segments (POS) and the basolateral membrane faces Bruch’s membrane, thereby separating the RPE from the choroid [2]. In a planar view, RPE cells exhibit a hexagonal shape. Microvilli extend from the apical surface of the RPE and envelop both rod and cone POS. The apical microvilli cloak a greater length of rods than cones [12]. They increase the RPE surface area 30-fold and promote a unique metabolic and functional relationship between the RPE cells and PRs, which is considered to be critical for the maintenance of visual function [1,13].

Laterally, the RPE membrane is the site of cell communication and adhesion. Apically, the composite of gap junctions, adherens junctions (AJs), and TJs form a physical barrier and are involved in the maintenance of cell polarity and the prevention of intramembranous diffusion between the basolateral and apical membrane domains [4,14]. Unlike most epithelia however, the RPE is unusual as its apical surface abuts solid tissue rather than a lumen. This variance allows the neural retina to influence epithelial polarity and the structure and function of TJs [4].

The infoldings of the RPE basal plasma membrane is typical of cells adept in transport. Intracellularly, the organelles tend to display domain specific distribution, with melanosomes primarily in the apical cytoplasm and the mitochondria, Golgi, and nucleus located basally [13].

The RPE has many functions that are vital to the maintenance of normal ocular function and homeostasis. These include phagocytosis of shed POS, retinoid conversion and storage, absorption of scattered light, RPE to PR apposition, and ion and fluid transport [13]. With such a broad spectrum of essential activities involved in vision, it is clear that primary dysregulation of these cells can lead to serious clinical effects, and indeed, given its interaction in many pathways, they must also be vulnerable to secondary damage.

The role of RPE junctional complexes in preventing paracellular movement of molecules and ions make RPE cells the protectors and key effectors of the oBRB [4]. However, they do not form an absolute and persistent wall. In fact, the RPE is moderately leaky in order to support the unique needs of the PRs. This is demonstrated by assessing transepithelial electrical resistance (TEER), which is a measure of paracellular ion movement. By inhibiting the passage of certain ions, TJs allow the RPE to regulate transport and establish concentration gradients between the neural retina and Bruch’s membrane [4].

In the context of the oBRB, TJs establish a barrier between the sub-retinal space and the choriocapillaris. RPE cells are responsible for the movement of nutrients and metabolic end products and serve to maintain the ion balance in the sub-retinal space [13]. Channels and pumps mediate the movement of ions and molecules via the transcellular pathway. In one direction, the RPE transports electrolytes and water from the sub-retinal space to the choroid, and in the other direction, the RPE transports glucose and other nutrients from the blood to the PRs [2]. Paracellular resistance is significantly higher than transcellular resistance, subsequently, water transport occurs mainly through transcellular pathways mediated by aquaporin-1 and -4 [2]. The Na^+^/K^+^-ATPase, which is located apically in RPE cells, provides the energy for the transport of electrolytes and water from the sub retinal space to the choroid [13]. A substantial amount of water is produced due to the large metabolic turnover of the retina. This constant elimination of water from the inner retina to the choroid produces an elimination force between the retina and RPE. This force is lost by the inhibition of the Na^+^/K^+^-ATPase. Proper anatomical apposition between RPE and PR cells is essential to the optics of the eye and health of PR cells [13].

Due to exposure to an intense level of light, toxic substances accumulate in PR cells daily [15]. In order for PRs to carry out their role of light transduction efficiently, the POS undergo a constant renewal process [15]. Each rod regenerates its outer segment in 7–12 days [13]. New POS are built at the cilium from the base of the outer segments. The highest concentration of radicals, photo damaged proteins, and lipids are located in the tips, which are shed from the PRs. Shed POS are phagocytosed by the RPE and the process of disk shedding and phagocytosis is tightly coordinated between the RPE and PR [15]. Through this coordinated POS tip shedding and the formation of new POS, a constant length of POS is maintained [15]. Lipofuscin compounds accumulate in the RPE as a consequence of the cells’ role in phagocytosing the POS membrane [13]. The RPE facilitates the recycling of the digested shed POS, allowing for essential molecules to be recycled to PRs [2,15]. This process of POS shedding is under circadian control, with the major burst of phagocytosis taking place with the onset of light [16].

The RPE also produces a number of growth factors and other soluble proteins that are essential for the maintenance and structure of the retina and the choriocapillaris [6]. Vascular endothelial growth factor (VEGF) is secreted in low concentrations at the basolateral surface of the RPE in the healthy eye where it prevents endothelial cell apoptosis and is essential for maintaining an intact endothelium associated with the choriocapillaris. It is thought to act as a permeability factor stabilizing the fenestrations of the endothelium [2,13].

## 3. Tight Junctions (TJs)

TJs are composed of transmembrane proteins and peripheral membrane proteins that interact with each other to form a complex network. Transmembrane proteins extend into the paracellular space, creating a seal characteristic of TJs [3]. These proteins include the MARVEL and claudin family members and JAMs. Scaffolding proteins including ZO-1, ZO-2, and ZO-3 bind to transmembrane proteins, linking these to the cytoskeleton [14]. The TJs create a barrier to paracellular diffusion of solutes as well as the maintenance of cell polarity between the basolateral and apical plasma membrane domains, which are often referred to as the “barrier” and “fence” function, respectively [17].

Since their identification via electron microscopy in 1963, we have learnt a great deal more about the location and composition of TJs [18]. In ultra-thin sections of brain tissues, TJs appear like a sequence of fusions (or kisses), which are formed between two adjacent cells by the outer leaflets of the plasma membrane. At higher magnification, it becomes clear that the membranes are not fused, but rather, are in tight contact with each other [14].

Secretions of the neural retina regulate the assembly, maturation, and tissue-specific properties of these TJs [4]. Initially, adhesive membrane proteins of AJs and TJs form adhesion complexes at sites of cell-to-cell contact. Subsequently, they organise into zipper like structures by lateral adhesion along the cell border [7]. The intracellular partners of transmembrane adhesive proteins also vary during junction maturation and stabilization. Adhesions are in dynamic equilibrium, even after stable contacts have been formed and recycle continuously between the plasma membrane and cytoplasm [8]. In epithelial cells, TJs and AJs follow a well-defined spatial distribution along the cellular cleft with TJs located at the most apical regions and AJs below them [8]. TJs make the most significant contribution to the paracellular component of TEER [4].

The TAMP (TJ-associated MARVEL protein) family members include occludin (MarvelD1), tricellulin (MarvelD2), and MarvelD3 [19,20]. They contain a conserved four-transmembrane MARVEL (Mal and related proteins for vesicle trafficking and membrane link) domain [21]. Occludin, tricellulin, and marvelD3 have both redundant and unique contributions to epithelial function [19,20]. Occludin was the first transmembrane protein identified in the TJ. It has a molecular mass of 65 kDa, with two extracellular loops, aforementioned four transmembrane loops and its amino and carboxy termini localised intracytoplasmically [22]. Occludin binds directly with the three ZO proteins [8,23] and these interactions are required for the TJ localisation of occludin [20]. Occludin’s two extracellular loops are involved in cell–cell adhesion and regulating paracellular permeability, respectively [3]. The C-terminal interacts with ZO-1 and ZO-2 and the last 150 amino acids interact with F-actin [3,22,23]. Expression of C-terminally truncated occludin in MDCKII cells resulted in increased paracellular permeability [24,25]. Over-expression of chicken occludin in MDCK cells leads to an increase in TEER, and therefore a decrease in paracellular permeability [3,26]. In combination, this suggests that occludin plays a potentially pivotal role in established paracellular permeability.

Occludin plays an essential regulatory role in the function of TJs, however, it is unnecessary for TJ formation [27,28,29,30]. Occludin associates with ZO-1 at the TJ upon recruitment by JAM-A [7]. This interaction is essential in modulating the function of occludin at the TJ [3]. Occludin has a half-life of approximately 1.5 h and rapidly dissociates from the TJ, indicating that it may facilitate TJs in adapting rapidly to physiologic changes [30,31]. Phosphorylation of occludin has been shown to act as an important regulatory mechanism. Post-translational phosphorylation status of occludin has been found to influence its location within the TJ and its regulation of paracellular permeability [26,32,33]. Threonine and serine phosphorylation of occludin appears to occur in conjunction with the preservation of TJ integrity [3]. Conversely, tyrosine phosphorylation of occludin appears to disrupt the association of ZO-1 and occludin, and demonstrates increased paracellular permeability [3]. In phosphorylated occludin, the binding of Z0-1, Z0-2, and Z0-3 to the C-terminal tail were decreased in comparison to non-phosphorylated occludin [3,34].

The Claudins are a family of transmembrane proteins of which there are more than 24 established members [14]. Like occludin, claudins have four membrane spanning regions, two extracellular loops, and two cytoplasmic termini/intracellular domains [14]. C-terminal amino acids encode PDZ-binding motifs, which are highly conserved throughout the claudin family. Through the C-terminal domain, claudins directly interact with peripheral PDZ-domain containing proteins including- ZO-1, ZO-2, and ZO-3 [14]. The first extracellular domain determines TEER and paracellular charge selectivity [35].

Claudin proteins mediate robust cell–cell adhesion and directly regulate permeability and selectivity [14,36]. They are almost certainly the main proteins important for TJ strand formation [14,30]. Claudins form pores facilitating the passive diffusion of molecules through the paracellular space [37]. Each claudin is thought to have a unique effect on selectivity and permeability [28,37] with the specific paracellular properties of different epithelia resulting from their individual pattern of claudin expression [36]. In the RPE, the claudins that are expressed vary among species [30]. In humans, claudin-3, claudin-10, and claudin-19 were detected in TJs [10] whereas in chicks, claudin-19 is not expressed and claudin-20 is the major claudin [9]. Some claudins have a limited distribution expressed in a tissue-specific manner, for example, claudin-5 appears to be confined to endothelial cells [8,38]. Claudin-1 to claudin-8 bind directly with ZO-1, ZO-2, and ZO-3 [8,39] via the cytoplasmic domain of claudins and the first PDZ domain of ZO proteins [39].

Claudin-19 is expressed and enriched in the RPE, where it is by far the predominant claudin [9]. High levels of claudin-19 are also found in the kidneys [9]. Claudin-19 determines permeability and semi-selectivity of the TJs in the RPE. By knocking-down claudin-19, the expressed claudin-3 was inadequate to form effective TJs. In contrast, the knockdown of claudin-3 demonstrated no effects [9]. Mutated claudin-19 affects multiple stages of RPE and retinal differentiation through its effects on multiple functions of the RPE [40,41]. Of note, claudin-10 was only expressed in a subset of cells [10].

The JAMs are a family of transmembrane proteins, of which four members have been identified: JAM-A, JAM-B, JAM-C, and JAM4/JAML [14]. JAMs are known to interact with many other proteins and may modulate TJ function by targeting other proteins to the TJ [8,14]. JAM-A can co-localize with occludin, ZO-1, and cingulin [7]. JAMs are members of the immunoglobulin superfamily and are expressed in epithelial cells along with endothelial cells, platelets, and leukocytes [14]. They are composed of a single transmembrane domain, and an extracellular domain containing two Ig-like motifs [14]. A PDZ-binding motif at the C terminus appears to be involved with the mediation of interaction with TJ scaffolding proteins, which appears to be important for the proper function of the TJ [42].

Peripheral membrane proteins anchor transmembrane proteins to the actin cytoskeleton and allow them to organize in the membrane and initiate cell signalling [3,14]. These include ZO-1, ZO-2, ZO-3, and cingulin [7].

The ZO proteins belong to the family of membrane associated guanylate kinases (MAGUK) that possesses a distinct molecular organisation. The core structure is composed of one or more PDZ domains, a Src homology 3 (SH3) domain, and a guanylate kinase (GUK) domain [14,43]. ZO-1 was the first TJ protein discovered in both epithelial and endothelial cells [44,45]. ZO-2 and ZO-3 were later discovered to localise to TJs with a similar sequence homology to ZO-1 [45]. ZO-1 has been shown to play a central role in the assembly and function of TJs [14]. PDZ domains facilitate the formation of specific multi-protein complexes including those necessary for TJ formation by recruiting downstream proteins in a signalling pathway [3]. At the TJ, the PDZ domain binds to the actin cytoskeleton through the C-terminal end and forms a bridge between the C-terminal sequences of occludin and β-actin [3]. It interacts with ZO-2 and ZO-3, its binding partners, through its second PDZ domain [11,43]. ZO proteins form a complex with AJ proteins in non-polarised cells where TJs have not formed, but upon polarisation, ZO proteins can separate from the AJ and concentrate in the TJ where they interact with TJ proteins such as claudins and occludin [46]. ZO-1 has also been found to participate in the regulation of gene expression via its binding in the nucleus to a transcription factor called Zonula-Occludens Associated Nucleic Acid-Binding protein (ZONAB). This interaction may allow TJs to regulate epithelial cell proliferation and cell density [47,48].

ZO-1 has been shown to be present in many ocular tissues including several retinal layers in the mammalian retina. This displays the role of ZO-1 in a wide portfolio of cellular functions in addition to mediating barrier function [3].

## 4. Role in Pathology

As we have established, the RPE has a wide ranging and diverse number of functions and the oBRB plays a crucial role in establishing the optimal conditions for the functioning of PRs and as such, also for visual function. In this section, we will discuss how the disruption of the oBRB contributes to the pathogenesis of a range of ocular pathologies namely DR, and diabetic macular oedema (DMO), AMD, CSCR, Sorsby’s fundal dystrophy, and RP, and also examine conditions associated with mutations in CLDN-19. It should be noted that this is not a complete list of all ocular conditions caused by the disruption of the oBRB, however, for the purpose of the review, we will focus solely on these conditions.

## 5. Diabetic Retinopathy

Diabetic retinopathy (DR) is the leading cause of blindness among working aged individuals in developed countries. In type 1 diabetes, proliferative diabetic retinopathy (PDR) is the most common sight threatening lesion and is characterised by neovascularisation secondary to a hypoxic insult [2]. In type 2 diabetes, DMO is the primary cause of visual loss, resulting from vascular leakage due to the breakdown of the iBRB [2]. It has been well described previously that iBRB alteration from endothelial cell dysfunction leads to DMO and subsequently retinopathy progression [1]. However, less is known about the role of the oBRB in DR.

There is growing evidence for diabetes induced oBRB dysfunction as observed in both human and animal studies [49,50]. The oBRB specific leakage has been visualised in diabetic and ischemic rodents demonstrating diabetic and ischemia-induced breakdown of TJs in the RPE, with one study reporting that the oBRB contributed a third of the total vascular leakage in diabetic retinas [51]. Significant depletion of occludin in the RPE of ischemic and diabetic rodents was also observed, suggesting that this leakage is a consequence of the breakdown of TJs in the oBRB [51]. In rodents with early stages of diabetes (those without discernible vascular leakage or TJ breakdown), reduced RPE absorptive capacity occurred before the breakdown of TJ strands [52]. In examining ZO-1 staining in hyperglycaemic rats, the TJ appeared wider and demonstrated small holes [52]. It was found that as a consequence of these conditions, the retina becomes highly hypoxic, which causes the upregulation of hypoxia inducible factor-1 (HIF) alpha and VEGF [39]. Upregulation of the VEGF signalling pathway ultimately leads to the loss of barrier function and TJ-integrity in the oBRB [51,52]. The oBRB breakdown then results in the leakage of blood contents and influx of osmolytes, which precedes subretinal oedema and exudative retinal detachment [51]. Furthermore, hyperglycaemia induces a loss of Na^+^/K^+^-ATPase function, which could impair the transport of water from the sub retinal space to choriocapillaris and consequently might contribute to DMO development [2].

## 6. Age-Related Macular Degeneration (AMD)

Age-related macular degeneration (AMD) is the leading cause of central retinal vision loss in developed countries, with an estimated one in 10 people over the age of 55 showing signs of the condition [53]. It is a complex, multifactorial disease characterised by RPE dysfunction and macular PR loss [13]. Advanced AMD presents in two forms that are generally referred to as “dry” or “wet”. Choroidal neovascularisation (CNV) is associated with wet AMD and occurs when new blood vessels sprout from the underlying choroidal vasculature and disrupt the integrity of the retina, leading to acute visual loss [54,55]. Geographic atrophy (GA) secondary to dry AMD occurs when the RPE begins to degenerate in the region of the macula leading to cone PR cell death and eventual central vision loss [55,56].

As discussed, the basal surface of the RPE rests on Bruch’s membrane. In the aging eye, extracellular material is deposited in Bruch’s membrane [13] and as a result, the thickness of Bruch’s membrane increases and permeability decreases [56,57]. Aging involves an accumulation of oxidative insults and a concomitant decrease in protective mechanisms [58]. The accumulation of lipofuscin in the RPE has been suggested to act as a starting point [15]. The initial triggers for age-related degenerative diseases are thought to be oxidative damage [15,58]. In patients with AMD, the RPE’s adaptive response to stress becomes dysregulated and an increasing imbalance of protective and toxic factors contributes to macular damage and the development of retinal lesions [15]. Reduced capacity of the RPE to absorb light energy is also an important factor in the cascade of events leading to AMD [15].

VEGF is secreted from the basolateral side of healthy RPE cells and is involved in the regulation of choroidal vasculature. Overexpression of VEGF is an important factor in the pathogenesis of CNV in wet AMD [6]. An imbalance between pro-angiogenic and anti-angiogenic factor in favour of the formation of new vessels [59] is likely to occur as a result of local hypoxia, inflammation, and dysregulated wound healing secondary to RPE cell loss [15]. It is the integrity of the oBRB that keeps the choroidal vascular response from invading the retina and changing dry into wet AMD [1] and the loss of this RPE–RPE attachment can induce VEGF overexpression [60]. RPE degeneration, tears, drusen formation, or apoptosis associated with aging can cause a loss of the RPE–RPE attachment [60]. Intravitreal agents that block VEGF have revolutionised the care of patients with wet AMD, decreasing growth, and leakage from CNV lesions, and preventing moderate and severe visual loss [59].

## 7. Central Serous Chorioretinopathy (CSCR)

Central serous chorioretinopathy (CSCR) is a disease of the retina characterised by serous detachment of the neurosensory retina secondary to focal lesions of the RPE. The mechanism whereby sub retinal fluid is produced is poorly understood [61]. It is thought that the choroidal vascular hyperpermeability occurs in CSCR, possibly as a result of stasis, ischaemia, or inflammation. Once the hydrostatic pressure is sufficiently high, the RPE is pushed forward, leading to discontinuing of the barrier and promoting pigment epithelial detachments. This overwhelms the barrier function of the RPE and leads to fluid accumulation within the RPE [62].

## 8. Sorsby’s Fundus Dystrophy

Sorsby’s fundus dystrophy is an autosomal dominant condition of the central retina. Similar to AMD, there is an accumulation of drusen deposits in Bruch’s membrane [13]. It is caused by variants in the gene for tissue inhibitor of metalloproteinases-3 (TIMP-3), which has been found in the drusen-like deposits that occur in patients with the condition [63]. TIMP-3 functions to regulate the thickness of Bruch’s membrane by inhibiting matrix metalloproteinases and also inhibits VEGF signalling and therefore angiogenesis [63]. Sorsby’s fundus dystrophy leads to eventual CNV and PR cell loss [63].

## 9. Retinitis Pigmentosa

Retinitis Pigmentosa (RP) is the term given to a set of hereditary retinal diseases that feature degeneration of rod and cone PRs and characteristic retinal deposits [64]. In most cases, it is due to the primary degeneration of rods with secondary cone degeneration [65]. In the first instance, RP patients typically lose night vision, followed by peripheral vision, and eventually central vision later in life [64,66]. Rhodopsin was the first gene identified and encodes rod visual pigment. The gene products localised in rods being involved in several visual pathways including a protein of rod visual transduction, cytoskeleton proteins, trafficking proteins, and PR differentiation [65]. There are at least 200 different mutations in more than 60 genes that are linked to RP [67]. Many of the pathways vary dramatically ranging from visual pigments to visual cycle to metabolism and RNA splicing [65]. In RHO(-/-) knockout mice (mice that carry a targeted disruption of the rhodopsin gene and present with rapid PR degeneration) upregulation of ZO-1 has been demonstrated when compared to wild type [66]. This finding may represent a physiological response to PR cell degeneration and appears to be associated with the retinal vasculature of the RHO (−/−) mice. In RHO (−/−) mice, blood vessels appear more clustered and thicker in appearance compared to the wild-type [66]. In contrast, a study looking at retinal degeneration 9 (Rd9) mice (mice that represent a rare model of X-linked RP) demonstrate a decrease in the intensity of ZO-1 staining [67].

In addition, RP may be complicated by cystoid macular oedema (CMO), which represents an important and treatable cause of central visual loss in RP [68]. The underlying pathogenesis remains unclear, however, breakdown of the BRB is a proposed mechanism for the development of CMO in RP [68]. The release of toxic products from the degenerating retina or RPE may cause CMO by disrupting the BRB. The iBRB is the primary site of vascular leakage in RP complicated by CMO in both patients with early and severe RP. Interestingly, patients with more extensive RP demonstrated greater leakage through the RPE compared to those with less extensive RP [69]. In eyes with RP alone, albumin leakage was greatest from the iBRB, however, in RP-associated with other complications such as aphakia and glaucoma leakage varied between iBRB and oBRB [70].

### Mutations in CLDN-19

As discussed earlier, claudin-19 is the predominant claudin in the human RPE [9]. Mutations in CLDN-19 cause the renal disease familial hypomagnesemia with hypercalciuria and nephrocalcinosis (FHHNC); ocular involvement (FHHNCOI) results in a varied array of ocular defects demonstrating incomplete penetrance [40,41]. In severe cases, symptoms appear in infants as young as 4–9 months old including bilateral macular coloboma, chorioretinal degeneration, nystagmus, strabismus, and visual loss [41]. Another study has reported that patients affected this mutation present with disrupted optic disc development resulting in near blindness and horizontal nystagmus [40]. The pathogenesis of such defects in FHHNCOI is yet to be determined.

## 10. Conclusions

In conclusion, the RPE is a single layer of cells that separates the neural retina from the underlying Bruch’s membrane and choroid. The TJs between the RPE form the oBRB, which is essential for the maintenance of visual function. In this review, we discussed the RPE and the oBRB in detail and examined the effect of its dysfunction, which can lead to an array of pathological ocular conditions. Understanding the complex molecular function of the oBRB will inform our understanding of a range of retinal diseases and could ultimately lead to the development of therapies aimed at restoring oBRB function and integrity.

## Figures and Tables

**Figure 1 ijms-21-00211-f001:**
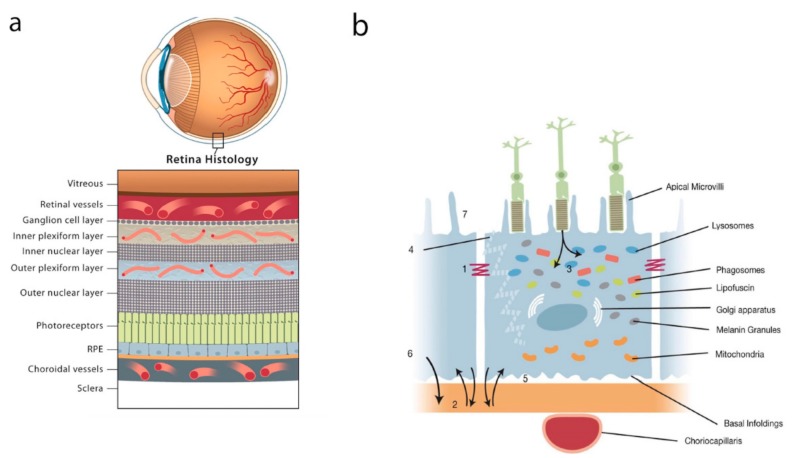
(**a**) Topographic relationship of the retinal pigment epithelium (RPE) within the retina. RPE anatomy and function. (**b**) The functions of the RPE are labelled as follows: (1) Tight junction (TJ) of the outer blood retina barrier (oBRB); (2) Transport of fluid, nutrients, and metabolites (paracellular and transcellular); (3) Phagocytosis of photoreceptor outer segments (POS); (4) Absorption of scattered light; (5) Retinal adhesion; (6) Paracrine secretion (including vascular endothelial growth factor (VEGF)); (7) Maintaining balance across the sub-retinal space.

**Table 1 ijms-21-00211-t001:** A summary of the key differences between the outer blood retina barrier (oBRB) and inner blood retinal barrier (iBRB).

Outer Blood Retinal Barrier	Inner Blood Retinal Barrier
Formed by tight junctions (TJ) between neighbouring retinal pigment epithelium (RPE) cells [1]. Rests on underlying Bruch’s membrane [1]	Formed by TJ between neighbouring retinal endothelial cells [1]. Rests on a basal lamina that is covered by the processes of astrocytes and Müller cells [1]
Regulates the paracellular movement of fluids and molecules between the choriocapillaris and the retina [1]	Regulates the paracellular movement of fluids and molecules across retinal capillaries [1]
Claudin-19 is the predominant claudin [9], claudin-3 and -10 are also expressed [10]	Claudin-5 is the most predominant claudin, claudin-1 and -2 are also expressed [11]
Plays a fundamental role in the microenvironment of the outer retina [1] including regulating access of nutrients from blood to photoreceptors (PRs), eliminating waste products, and maintaining retinal adhesion [1]	Plays a fundamental role in the microenvironment of the neural retina [1]
The relationship between the RPE apical villi and PR is considered to be crucial in maintaining visual function [1]	Regulatory signals of the retinal neuronal circuitry are transmitted by astrocytes, muller cells and pericytes thereby influencing the activity of the iBRB [1]

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
