# Peer review of "Tight Junctions of the Outer Blood Retina Barrier"

_ijms, 2019, doi:10.3390/ijms21010211_

Round 1
Reviewer 1 Report
Tight junctions of the outer blood retinal barrier
Aisling Naylor, Alan Hopkins, Natalie Hudson and Matthew Campbell
The authors entitled their review “Tight junctions of the outer blood retinal barrier”, and they organized this review in three parts, (i) the RPE, (ii) tight junctions, and (iii) pathology, where they describe how the disruption of the oBRB contributes to the pathogenesis of different ocular pathologies.
In the first part, they describe the function of the RPE and in this context they shortly mention the TJ as barrier between the sub-retinal space and the choriocapillaris.
The second part is a review about the TJ,and especially about TJ proteins. The only link to the RPE in this part is claudin-19 and mutations in this claudin in many ocular
abnormalities, including macular colobomata.
Line 175: Claudin-19 is expressed and enriched in the RPE, where it is by far the predominant claudin and the only claudin that is RPE specific [27]….
Claudin-19 is also expressed in the kidney, it is not RPE specific.
The claudin-19 mutation should be described under pathologies.
Furthermore, there are other claudins and TJ proteins detected in the RPE and should be mentioned in this review.
In the last part, the authors describe of different ocular pathologies, namely DR, and diabetic macular oedema (DMO), AMD, CSCR, Sorsby’s fundal dystrophy and RP. In this part, they mention a depletion of occludin in DR and an upregulation of ZO-1 in an animal model of autosomal recessive RP, in some of the other pathologies, they mention an involvement of the oBRB without providing any detailed information.
In summary, this review contains little information about TJs of the outer blood retinal barrier, it provides some information about the function of the RPE, about TJs and about different ocular pathologies with involvement of the RPE. There are different studies on TJ proteins in the RPE, on animal models with disturbed BRB, and a good review should include these information.
Minor points:
Line 59: The RPE is composed of a single layer of cells joined laterally towards their apices by TJs between adjacent cell walls.
RPE cells don’t have cell walls, please write cell membranes.
Line 99: …of the Na+/K+-ATPase – Please write Na+/K+-ATPase
Line 110: ………..recycled to PRs [2, 12] This……………….
Please insert a dot after [2, 12]
Line 122: These proteins include occludin, claudin family members and JAMs.
Please mention not only occluding, but the TJ-associated MARVEL proteins (TAMP), comprising occludin, tricellulin, and MarvelD3.
Line 306: The TJs between the RPE form the OBRB – please write oBRB
Author Response
Line 175: Claudin-19 is expressed and enriched in the RPE, where it is by far the predominant claudin and the only claudin that is RPE specific [27]….
Claudin-19 is also expressed in the kidney, it is not RPE specific.
The claudin-19 mutation should be described under pathologies.
This has been amended.
Furthermore, there are other claudins and TJ proteins detected in the RPE and should be mentioned in this review.
This has been amended.
In the last part, the authors describe of different ocular pathologies, namely DR, and diabetic macular oedema (DMO), AMD, CSCR, Sorsby’s fundal dystrophy and RP. In this part, they mention a depletion of occludin in DR and an upregulation of ZO-1 in an animal model of autosomal recessive RP, in some of the other pathologies, they mention an involvement of the oBRB without providing any detailed information.
We have elaborated on this section in more detail.
Minor points:
Line 59: The RPE is composed of a single layer of cells joined laterally towards their apices by TJs between adjacent cell walls.
RPE cells don’t have cell walls, please write cell membranes.
Amended.
Line 99: …of the Na+/K+-ATPase – Please write Na+/K+-ATPase
Amended.
Line 110: ………..recycled to PRs [2, 12] This……………….
Please insert a dot after [2, 12]
Amended.
Line 122: These proteins include occludin, claudin family members and JAMs.
Please mention not only occluding, but the TJ-associated MARVEL proteins (TAMP), comprising occludin, tricellulin, and MarvelD3.
Amended.
Line 306: The TJs between the RPE form the OBRB – please write oBRB
Amended.
Reviewer 2 Report
The manuscript by Aisling Naylor et al. is a concise review on the tight junctions of the outer retinal barrier. The review is well written and contains relevant information about the specific aspects of retinal pigment epithelial cells including pathological conditions.
Inclusion of a figure depicting the local cellular arrangement and a table summarizing the specific aspects of the outer retinal barrier tight junctions would greatly improve the understandability of the manuscript.
Author Response
Figures now included.
Round 2
Reviewer 1 Report
In the revised version, the authors addressed most of the reviewers suggestions.
Minor points:
Line 256: ….occurred occurred Please delete occurred
Line 257: ….. Zo-1 Please write ZO-1
Line 258: ….[51].It Please insert a space
Line 290: …. changing dry into wet Please write ….dry into wet AMD
Author Response
Amended